# Platelet-Rich Plasma in Alopecia Areata—A Steroid-Free Treatment Modality: A Systematic Review and Meta-Analysis of Randomized Clinical Trials

**DOI:** 10.3390/biomedicines10081829

**Published:** 2022-07-29

**Authors:** Fanni Adél Meznerics, Kata Illés, Fanni Dembrovszky, Péter Fehérvári, Lajos Vince Kemény, Kata Dorottya Kovács, Norbert Miklós Wikonkál, Dezső Csupor, Péter Hegyi, András Bánvölgyi

**Affiliations:** 1Department of Dermatology, Venereology and Dermatooncology, Faculty of Medicine, Semmelweis University, 1085 Budapest, Hungary; meznerics.fanni@stud.semmelweis.hu (F.A.M.); kemeny.lajos@med.semmelweis-univ.hu (L.V.K.); katiedorotie@gmail.com (K.D.K.); wikonkal.norbert@med.semmelweis-univ.hu (N.M.W.); 2Centre for Translational Medicine, Semmelweis University, 1085 Budapest, Hungary; illes.kata@bajcsy.hu (K.I.); dembrovszky.fanni@semmelweis-univ.hu (F.D.); fehervari.peter@univet.hu (P.F.); csupor.dezso@pharmacognosy.hu (D.C.); p.hegyi@tm-centre.org (P.H.); 3Department of Otorhinolaryngology, Head and Neck Surgery, Bajcsy-Zsilinszky Hospital, 1106 Budapest, Hungary; 4Institute for Translational Medicine, Medical School, Szentágothai Research Centre, University of Pécs, 7624 Pécs, Hungary; 5Department of Biomathematics and Informatics, University of Veterinary Medicine, 1078 Budapest, Hungary; 6Institute of Clinical Pharmacy, Faculty of Pharmacy, University of Szeged, 6720 Szeged, Hungary; 7Division of Pancreatic Diseases, Heart and Vascular Center, Semmelweis University, 1085 Budapest, Hungary

**Keywords:** alopecia areata, patchy alopecia, platelet-rich plasma, treatment, topical

## Abstract

Background: Alopecia areata (AA) is a chronic autoimmune condition that can lead to a serious deterioration in patients’ quality of life. The first line of treatment in patchy AA is triamcinolone acetonide (TrA); however, the efficacy of the treatment varies greatly. Our aim was to investigate the therapeutic effects of platelet-rich plasma (PRP) in the treatment of AA. Method: We performed a systematic literature search in four databases. Randomized clinical trials (RCT) reporting on patients with AA treated with PRP were included, comparing PRP with TrA or a placebo. The primary outcome was the Severity of Alopecia Tool (SALT) score. Results: Our systematic search provided a total of 2747 articles. We identified four studies eligible for quantitative analysis. The pooled mean differences from the four studies did not exhibit a significant difference in the mean change in the SALT score when PRP and TrA groups were compared (MD =−2.04, CI: −4.72–0.65; I^2^ = 80.4%, *p* = 0.14). Conclusions: PRP is a promising topical, steroid-free treatment modality in the therapy of AA. No significant difference was found between PRP and TrA treatment; however, further high-quality RCTs are needed to further assess the efficacy of PRP treatment and strengthen the quality of evidence.

## 1. Introduction

Alopecia areata (AA) is an autoimmune condition characterized by inflammation-induced hair loss due to the collapse of the hair follicles’ immune privilege [1]. It can affect the scalp, the beard, or even the whole body, leading to a serious deterioration in patients’ quality of life [2]. In the case of AA of the scalp, three categories can be differentiated based on the extent of the disease-affected area: limited patchy AA with less than 50% scalp involvement, extensive patchy AA with more than 50% scalp involvement, and alopecia totalis, affecting the whole scalp [3].

A wide spectrum of topical and systemic agents is used in the management of AA; however, there is a lack of consensus on a standard treatment modality due to the disease’s varying response to therapy [4]. The limited patchy forms of AA are usually treated with topical agents, such as corticosteroids, contact immunotherapy (1-chloro,2,4-dinitrobenzene (DNCB), squaric acid dibutylester (SADBE), and 2,3-diphenylcyclopropenone (DPCP)), and minoxidil; however, the efficacy of the latter treatments is questionable [5]. According to guidelines, the first line of treatment in limited patchy AA is triamcinolone acetonide (TrA) administered intralesionally [5,6]. Besides the frequently disputed efficacy of TrA treatment, side effects, such as skin atrophy, teleangiectasia, and hypopigmentation, frequently occur. Additionally, the use of steroids is alarming to many; a phenomenon called steroid phobia exists [7]. These provide additional reasons to look for new topical, steroid-free treatment modalities.

Platelet-rich plasma (PRP) is a relatively new, presently evolving treatment modality that is playing an increasingly important role in the field of dermatology. The efficacy of PRP varies greatly and is being investigated in numerous dermatological disorders, such as androgenic alopecia, acne scar treatment, or chronic wound management [8,9,10]. PRP is prepared from whole blood by a centrifugation process to achieve a product that is rich in platelets, growth factors, and cytokines. Based on the number of centrifugations, single-spin and double-spin preparation methods can be differentiated [11,12]. PRP was shown to stimulate cell proliferation in the dermal papilla (DP), increase DP cell survival through antiapoptotic effects, and stimulate hair regrowth by prolonging the anagen phase of the hair cycle [13].

The Severity of Alopecia Tool (SALT) score, the most widely used method to monitor the response to therapy, is an objective outcome measure to evaluate the severity of the disease. It is determined by visually assessing the percentage of hair loss, resulting in a score from 0 to 100 analogous to the percentage of the affected area [14]. In limited patchy AA, the SALT score is lower than 50, indicating that the scalp involvement is below 50% [3,14].

Several studies have reported promising results of PRP in the treatment of AA [15,16,17,18,19,20], but there has been no systematic evaluation of randomized trials reporting on the therapeutic effect of PRP on AA to date. Our aim was to investigate the therapeutic effects of PRP in the treatment of limited patchy AA by conducting a systematic review and meta-analysis, comparing PRP with the first line of treatment in limited patchy AA, TrA [10,13]. Our hypothesis was that PRP is as good as TrA in the treatment of AA.

## 2. Materials and Methods

Our systematic review and meta-analysis are reported according to the PRISMA (Preferred Reporting Items for Systematic Reviews and Meta-Analyses) 2020 Statement [21]. This study was performed following the Cochrane Handbook’s recommendations for Systematic Reviews of Interventions Version 6.3. [22]. The review protocol was registered on PROSPERO (York, UK) under registration number CRD42021282807 (see https://www.crd.york.ac.uk/prospero, accessed on 14 October 2021); no amendments to the information provided at registration were made.

### 2.1. Literature Search and Eligibility Criteria

We performed a systematic literature search in four medical databases: MEDLINE (via PubMed), Cochrane Library (CENTRAL), Embase, and Web of Science, from inception to 15 October 2021. We applied the query ((alopecia areata OR patchy alopecia) AND (platelet rich plasma OR PRP OR steroid OR corticosteroid OR triamcinolone)) to all fields in the search engines. No language or other restrictions were imposed.

Randomized clinical trials (RCTs) reporting on patients with AA treated with PRP were included, comparing PRP with TrA or a placebo. The following population–intervention–control–outcome (PICO) framework was used:

P—Adult patients with patchy AA;

I—Intralesional autologous PRP injections to the AA-affected areas;

C_1_—Intralesional placebo injections to the AA-affected areas;

C_2_—Intralesional (TrA) injections to the AA-affected areas;

O—Primary outcome: SALT score; secondary outcomes: hair dystrophy, patient safety, cytokine expression, burning/itching sensation, Hair Regrowth Grade scale, Patient Global Assessment score, cell proliferation.

Publications without separate intervention and control groups (split scalp studies) were excluded.

### 2.2. Study Selection and Data Collection

We used EndNote X9 (Clarivate Analytics, Philadelphia, PA, USA) for the articles’ selection. Two independent authors (F.A.M. and K.I.) screened the publications separately for the title, abstract, and full text, and disagreements were resolved by a third author (F.D.). 

Two authors (F.A.M. and K.D.K.) independently extracted data into a predefined Excel spreadsheet (Office 365, Microsoft, Redmond, WA, USA). The following data were collected from each eligible article: first author, year of publication, study type, study location, number of centers included in the study, study design, demographic data (sample sizes, age, and percentage of participating females), details of the received treatments, and data regarding our outcomes (baseline SALT score and SALT score after treatment, and secondary outcomes) for statistical analysis. A third reviewer (F.D.) resolved the discrepancies. Based on the baseline SALT scores and the post-treatment SALT scores reported in the included studies, we calculated the mean change in the SALT score in both the PRP and TrA groups, and the mean difference (MD) between the two groups.

### 2.3. Quality Assessment and Quality of Evidence

The quality assessment of the outcomes was carried out separately by two reviewers (F.A.M. and K.D.K.) using the revised tool for assessing the risk of bias (RoB 2) [23]. Disagreements were resolved by a third reviewer (F.D.). The recommendations of the “Grades of Recommendation, Assessment, Development, and Evaluation (GRADE)” workgroup were followed to evaluate the quality of evidence [24]. 

### 2.4. Data Synthesis and Analysis

The mean baseline SALT score and SALT score after treatment were extracted for both the TrA and PRP groups. We calculated the change in the mean SALT scores for each study, and, where applicable, we also extracted the standard deviation (SD) of within-group differences. If this latter value was not published, but t-tests or ANOVA were used, we calculated the SD from the reported *p* values, or in the case of non-parametric tests, we obtained a conservative estimate of the SDs by adding together the reported before- and after-treatment SDs. Meta-analysis using random effects (DerSimonian and Laird) was performed following the recommendations of the Cochrane Handbook and Harrer et al. using R version 6.3 [22,25,26,27]. The heterogeneity of the studies was assessed with the Cochran Q test, with a significance level of 0.05 and I^2^ statistic, and forest plots were constructed.

## 3. Results

### 3.1. Search and Selection

Our systematic search provided a total of 2747 articles; after duplicate removal, we screened 2002 duplicate-free articles. After the title, abstract, and full-text selection, we identified six RCTs matching our PICO framework [15,16,17,18,19,20]; of these articles, we could use four RCTs for our quantitative synthesis [15,17,18,19]. The results of the other two articles [16,20] and the results of two additional articles, which also included a placebo as a comparator [18,20], are discussed in the systematic review with the secondary outcomes. The summary of the selection process is shown in Figure 1.

### 3.2. Main Characteristics of the Included Studies

The characteristics of the identified RCTs for the systematic review and meta-analysis are detailed in Table 1 and Table 2.

### 3.3. Primary Outcome (SALT score)

#### 3.3.1. PRP Compared to Triamcinolone Acetonide

Two studies evaluated the post-treatment SALT score 12 weeks after the first treatment session [15,17], one study 16 weeks after the first treatment session [18], and one at multiple timepoints: weeks 3, 6, 9, 12, and 24 [19] (see Table 1). We used the SALT score of the 12th-week evaluation of this study for our meta-analytical calculations. The pooled MDs from four RCTs with a total of 201 subjects did not show a significant difference in the mean change in the SALT scores between the PRP and TrA groups (MD = −2.04; CI: −4.72–0.65; I^2^ = 80.4%; *p* = 0.14) (see Figure 2). 

Due to the high heterogeneity, we performed a leave-one-out analysis; the results are detailed in Table 3. 

All studies included in our systematic review and meta-analysis showed a significant decrease in the SALT scores for PRP and TrA groups [15,16,17,18,19,20].

Two of the six publications originally selected used the SALT score as a primary outcome; however, due to missing data, we could not include them in our meta-analysis. 

Balakrishnan et al. found no statistically significant difference between the SALT scores of the two groups when compared at different timepoints; however, the response in the PRP group was better than that in the TrA group [16]. When the decrease in the SALT score at different timepoints was investigated, there was a significant difference in the decrease in the SALT score between the PRP group and the TrA group at the second evaluation, 4 weeks after the first treatment session (*p* = 0.028) [16]. After the last evaluation (12 weeks after the first treatment session), there was no statistically significant difference between the two groups [16]. Trink et al. found that the SALT score decreased significantly in the PRP group compared with the TrA group (*p* < 0.001) [20]. 

The number of patients with complete hair regrowth is detailed below, since we could not conduct a meta-analysis due to the highly variable evaluation timepoints of the studies.

Trink et al., found that, 12 months after the first treatment session, complete remission was achieved in 26.6% of the patients in the TrA group and 60.0% of the patients in the PRP group [20]. Albalat et al. reported 16 patients (40.0%) in the TrA group and 18 patients (45.0%) in the PRP group with complete remission 8 weeks after the first treatment session [15]. Hegde et al. found that 10 patients (40.0%) in the TrA group and 11 patients (44.0%) in the PRP group achieved nearly complete hair regrowth 5 months after the first treatment (*p* = 0.779) [18].

Two studies also used additional categorical scales to describe the decrease in the SALT score, both referring to them as the Hair Regrowth Grade (HRG) scale [16,19]. Comparing the HRG scale between PRP and TrA treatment, Balakrishnan et al. found no statistical significance, while Kapoor et al., reported a significant difference (*p* = 0.0002): more patients from the TrA group were in the grade IV (50–74% reduction in SALT score) and grade V (>74% reduction in SALT score) categories after treatment compared with the PRP group [16,19]. 

#### 3.3.2. PRP Compared to Placebo

Trink et al., found that PRP significantly increased hair regrowth in AA lesions compared with the placebo (*p* < 0.001). PRP treatment also led to increased hair regrowth when compared with the untreated side of the scalp (*p* < 0.001) [20].

Hegde et al. found that, 12 weeks after the first treatment session, PRP showed a higher percentage of regrowth than the placebo (*p* = 0.0108) [18].

### 3.4. Secondary Outcomes

#### 3.4.1. Patient Safety

##### Adverse Effects

Balakrishnan et al., Hegde et al., and Trink et al. did not record any side effects in the enrolled patients, nor were serious side effects observed in the study by Albalat et al. [15,16,18,20]. The only side effects observed in the study by Albalat et al. were erythema and a burning sensation after treatment sessions, without significant differences between the two groups [15]. Kapoor et al. reported atrophy in five patients from the TrA group, and the difference between the PRP and TrA groups was significant (*p* = 0.047) [19].

##### Administration-Related Pain

Three patients in the study by Balakrishnan et al. reported severe pain during administration in the PRP group, while there was none in the TrA group. Hegde et al. and Kapoor et al. also found significant differences between the two groups regarding pain (*p* < 0.05): significantly higher visual analog scale (VAS) scores were recorded and a higher number of patients reported pain in the PRP group (*p* < 0.0001) [18,19].

##### Recurrence Rates

Trink et al. reported a 38% relapse in the TrA group and no relapse in the PRP group 6 months after the first treatment. Twelve months after the first treatment, 71% of the patients in the TrA group experienced a recurrence of the disease, whereas this rate was 31% of the PRP-group patients [20]. At the follow-up visit 6 months after the first treatment, two patients (5%) in the PRP group and 10 patients (25%) in the TrA group reported recurrence, according to Albalat et al. [15].

Further secondary outcomes are detailed in the Supplementary Results in the Appendix A.

### 3.5. Risk of Bias Assessment

The results of the assessment of the risk of bias of the studies included in the meta-analysis and systematic review can be seen in Appendix A in the Appendix A. None of the studies included in the meta-analysis were at high risk of bias. In three articles, the randomization process [15,17,19], and in two articles, the measurement of the outcome [18,19], were ranked as “some concerns”. Deviation from the intended intervention, missing outcome data, and the selection of the reported results domains were at low risk of bias.

### 3.6. Quality of Evidence

The quality of evidence was low for the primary outcome (SALT score). 

## 4. Discussion

The studies included in our systematic review and meta-analysis all showed a significant decrease in the SALT score in both the PRP and TrA groups [15,16,17,18,19,20]. The pooled MDs from the four RCTs with a total of 201 subjects did not show a significant difference in the mean change in the SALT score between the PRP and TrA groups. Although we could not conduct a meta-analysis comparing PRP with a placebo, the included studies all concluded the superiority of PRP treatment [18,20]. These results could prove the efficacy of PRP as a steroid-free treatment modality. However, several factors could have influenced these results, such as the different doses of TrA used and the length of the follow-up.

The strength of the effect of TrA can be dose-dependent: RCTs investigating the most effective dilution of TrA showed that the 10 mg/mL dose achieved the best therapeutic response; however, due to the dose-dependent increasing risk of adverse effects, it is advised to start the treatment with lower doses [28,29]. Two of the four studies included in our meta-analysis used 5 mg/mL TrA, and two studies used 10 mg/mL TrA as a comparator [19]. The decrease in the SALT score was higher in the studies using a higher dose of TrA; nevertheless, one of the latter studies registered atrophy in five cases, assumably due to the higher doses of TrA. On the contrary, PRP can be used in an unlimited number of treatment sessions without increasing the risk of adverse effects [15,16,18,19,20]. 

Complete remission and recurrence are also important factors when choosing a therapy. Only one included RCT followed up patients for more than 6 months, and they recorded a higher number of cases with complete remission and lower recurrence rates with PRP compared with TrA one year after the first treatment session [20]. Albalat et al. followed up patients for 6 months, and they also recorded lower recurrence rates in the PRP group 6 months after the first treatment session [15].

Pain is also a considerable factor that can affect the utility of a therapeutic modality. Pain related to the treatment was more frequently reported in the PRP group; however, it could possibly be decreased with the application of a pre-treatment topical lidocaine–prilocaine cream with occlusion or with a minimally invasive application technique, such as microneedling [16,18,19,30].

### 4.1. Strengths and Limitations

Our systematic search was conducted only 6 months before the submission of this manuscript; therefore, it includes every relevant study regarding this topic. We implemented a rigorous selection protocol and only included RCTs to obtain the highest possible quality of evidence, and it is also notable that all included studies were published recently. To our knowledge, this is the first meta-analysis to date comparing the efficacy of PRP to TrA, the first line of treatment of limited patchy AA.

The two main limitations of this study are the small sample size and high heterogeneity. Besides the small sample size, the inclusion of the study of Kapoor et al. can also be a potential reason for the high heterogeneity. Kapoor et al. used higher doses of TrA, the treatment sessions were more frequent compared with the other studies, and the baseline SALT score was also higher in the TrA group than in the PRP group. These discrepancies could explain the significantly higher mean decrease in the SALT score in the TrA group (*p* < 0.0001). The leave-one-out analysis showed that, when excluding the study of Kapoor et al., I^2^ decreased to 9.0% and the confidence interval narrowed. Different preparation methods of PRP in the included studies could also lead to high heterogeneity, since the superiority of the double-spin method to the single-spin method was shown in previous studies [11,12].

### 4.2. Implications for Research

Our study demonstrated the efficacy of PRP in the treatment of patchy AA; however, further high-quality RCTs providing both within- and between-group detailed descriptive statistics are needed to better assess the efficacy and to strengthen the quality of evidence. The implementation of objective, comparable outcome measurements besides the SALT score could help to better assess complete remission, recurrence rates, and adverse effects. This would contribute to a better understanding of the pros and cons of each treatment modality and would also enable future systematic analysis using these parameters to further strengthen the quality of the current evidence.

Future RCTs should also focus on the comparison of PRP with different doses of TrA, since, while high doses of TrA lead to better improvement, they may also increase the risk of adverse effects [28,29]. A steroid-free treatment option, such as PRP, as a first-choice treatment could be beneficial, even if it shows slower improvement.

Follow-up protocols longer than 4 months would make it possible to see further differences between the two treatment modalities in order to better assess the complete remission and recurrence rates. 

### 4.3. Implications for Practice

Platelet-rich plasma is a steroid-free treatment modality that can be used in a virtually unlimited number of treatment sessions without increasing the risk of steroid-specific adverse effects [15,16,18,19,20]. The adverse effects of TrA treatment, such as atrophy, teleangiectasia, and hypopigmentation, can be especially problematic in the facial region. Since PRP is also safely used in facial rejuvenation, it could be an optimal therapeutic choice in facially localized AA [7,31,32]. The application of PRP with microneedling or fractional carbon dioxide laser treatment could be a more convenient method of administration, particularly in the facial region and in extensive cases of AA [30]. 

## 5. Conclusions

Platelet-rich plasma is a promising topical steroid-free treatment modality in the therapy of alopecia areata. No significant difference was found between PRP and TrA treatment; however, further high-quality RCTs are needed to better assess the efficacy of PRP and to strengthen the quality of evidence. PRP can be used in a virtually unlimited number of treatment sessions without increasing the risk of steroid-specific adverse effects, and it can also be an alternative option in the treatment of facially localized AA, in extensive cases of AA, or in cases of steroid phobia.

## Figures and Tables

**Figure 1 biomedicines-10-01829-f001:**
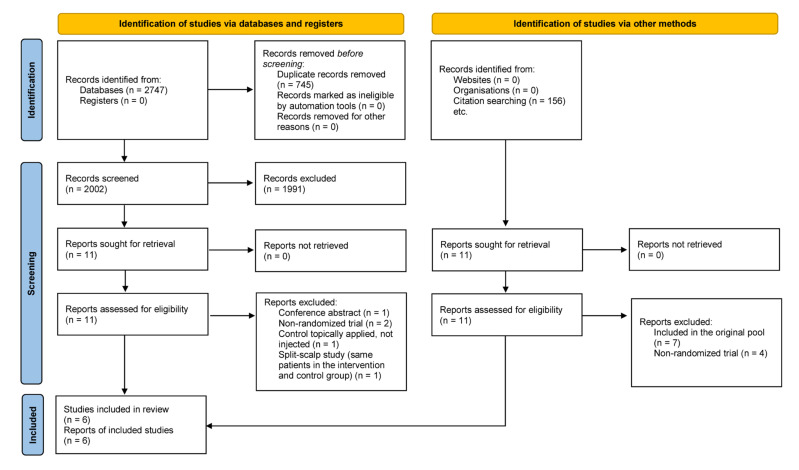
PRISMA flow diagram of the screening and selection process according to PRISMA 2020 guidelines [21].

**Figure 2 biomedicines-10-01829-f002:**
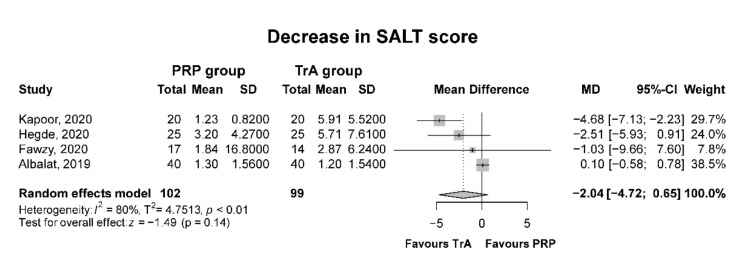
Forest plot for the mean decrease in the SALT score; platelet-rich plasma (PRP) compared to triamcinolone acetonide (TrA).

**Table 1 biomedicines-10-01829-t001:** Main characteristics of the included studies.

**First Author,** **Year of Publication**	**Country**	**Study Design**	**Number of Patients**	**Intervention**	**Control**	**Administration**	**Timepoints of Evaluation (weeks) _a_**
Studies included in meta-analysis							
Albalat, 2019 [15]	Egypt	RCT	80	PRP injection (double-spin method)	TrA injection (5 mg/mL)	3–5 sessions, 2-week intervals	12
Fawzy, 2020 [17]	Egypt	RCT	31	PRP injection (single-spin method)	TrA injection (5 mg/mL)	3 sessions, 4-week intervals	12
Hegde, 2020 [18]	India	RCT	50	PRP injection (double-spin method)	TrA injection (10 mg/mL), placebo	3 sessions, 4-week intervals	16
Kapoor, 2020 [19]	India	RCT	40	PRP injection (single-spin method)	TrA injection (10 mg/mL)	4 sessions, 3-week intervals	3, 6, 9, 12 _b_, 24
Studies included only in systematic review							
Balakrishnan, 2020 [16]	India	RCT	32	PRP injection (double-spin method)	TrA injection (10 mg/mL)	3 sessions, 4-week intervals	0, 4, 8, 12
Trink, 2013 [20]	Italy	RCT	30	PRP injection (single-spin method)	TrA injection (2,5 mg/mL), placebo	3 sessions, 4-week intervals	8, 24, 48

_a_ weeks after the first treatment session; _b_ timepoint used in our calculations. RCT: randomized clinical trial; PRP: platelet-rich plasma; TrA: triamcinolone acetonide.

**Table 2 biomedicines-10-01829-t002:** Patient characteristics of the studies included in the meta-analysis.

	Intervention (PRP) Group					Control (TrA) Group				
First Author, Year of Publication	Number of Patients	Age, Mean (SD)	Sex (Female % of Total)	Baseline SALT Score, Mean (SD)	Post-Treatment SALT Score, Mean (SD)	Numberof Patients	Age, Mean (SD)	Sex (Female % of Total)	Baseline SALT Score, Mean (SD)	Post-Treatment SALT Score, Mean (SD)
Albalat, 2019 [15]	40	30.8 (7.5)	15.0	1.7 (0.9)	0.4 (0.7)	40	36.3 (11.3)	15.0	1.7 (0.8)	0.5 (0.8)
Fawzy, 2020 [17]	17	31.4 (10.6)	23.5	5.6 (8.4)	3.8 (8.4)	14	34.2 (12.3)	28.6	4.2 (4.4)	1.4 (1.8)
Hegde, 2020 [18]	25	N/A	N/A	7.2 (3.8)	4.0 (5.3)	25	N/A	N/A	8.8 (5.8)	3.1 (5.1)
Kapoor, 2020 [19]	20	25.4 (4.9)	45.0	4.4 (2.5)	3.2 (2.0)	20	28.8 (8.6)	65.0	9.0 (1.4)	3.1 (0.8)

SD: standard deviation; N/A: data not available; PRP: platelet-rich plasma; TrA: triamcinolone acetonide; SALT score: Severity of Alopecia Tool score.

**Table 3 biomedicines-10-01829-t003:** Results of leave-one-out analysis.

	Quantitative Data Synthesis					Heterogeneity	
Study	PRP Group (*n*)	TrA Group (*n*)	Effect Size	95% CI	*p* Value	I^2^	T^2^
Overall Effect	102	99	−2.04	[−4.72; 0.65]	0.14	80.4%	2.24
Leave-one-out sensitivity analysis							
Albalat, 2019 [15]	62	59	−3.79	[−5.75; −1.83]	0.001	0.0%	0.04
Fawzy, 2020 [17]	85	85	−2.15	[−5.14; 0.83]	0.15	87.0%	5.57
Hegde, 2020 [18]	77	74	−1.94	[−5.62; 1.73]	0.30	85.0%	7.32
Kapoor, 2020 [19]	82	79	−0.57	[−2.54; 1.41]	0.57	9.0%	1.29

PRP: platelet-rich plasma; TrA: triamcinolone acetonide; CI: confidence interval.

## Data Availability

Not applicable.

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
