# Peer review of "Platelet-Rich Plasma in Alopecia Areata—A Steroid-Free Treatment Modality: A Systematic Review and Meta-Analysis of Randomized Clinical Trials"

_biomedicines, 2022, doi:10.3390/biomedicines10081829_

Round 1
Reviewer 1 Report
Very interesting Systematic literature search in four databases. Randomized clinical trials (RCT) reporting on patients with AA treated with PRP were included, comparing PRP with TrA or placebo. They concluded that PRP is a promising topical steroid-free treatment modality in the therapy of AA. Even thought only 4 clinical study could be confirm and one of them have a very small baseline SALT score.
in the introduction please be more precise knowing that AA is not only inflammation driven : Alopecia areata is an autoimmune disease resulting from a breach in the immune privilege of the hair follicles: the body attacks its own anagen hair follicles and suppresses or stops hair growth. T cell lymphocytes cluster around affected follicles that might cause at least inflammation and subsequent hair loss. Psychological stress and illness are possible factors in bringing on alopecia areata in individuals at risk.
The article bring conclusion with caution since only 4 clinical studies were compare.
Reviewer 2 Report
This is interesting study based on meta-analysis aiming to assess the effectiveness of platelet-rich plasma treatment in alopecia areata comparing to triamcinolone acetonide application. Introduction gives relevant background and methodology is properly described.
I have only a few minor suggestions for Authors:
1) Line 47: „2,4,dinitrobenzene” – it should be: : „2,4-dinitrobenzene”
2) Line 72: Is references [12-17] placed correctly? It should be probably in line 70: „in the treatment of AA [12-17]”
3) The quality of Figure 1 should be improved. The text under the diagram is illegible.
4) Some information on double-spin method and the single-spin method should be added
